# Signal Diversity for Laser-Doppler Vibrometers with Raw-Signal Combination

**DOI:** 10.3390/s21030998

**Published:** 2021-02-02

**Authors:** Marvin Schewe, Christian Rembe

**Affiliations:** Institute of Electrical Information Technology, Clausthal University of Technology, Leibnizstr. 28, D-38678 Clausthal-Zellerfeld, Germany; rembe@iei.tu-clausthal.de

**Keywords:** speckle, laser-doppler vibrometry, LDV, diversity reception, signal diversity, laser speckle

## Abstract

The intensity of the reflected measuring beam is greatly reduced for laser-Doppler vibrometer (LDV) measurements on rough surfaces since a considerable part of the light is scattered and cannot reach the photodetector (laser speckle effect). The low intensity of the reflected laser beam leads to a so-called signal dropout, which manifests as noise peaks in the demodulated velocity signal. In such cases, no light reaches the detector at a specific time and, therefore, no signal can be detected. Consequently, the overall quality of the signal decreases significantly. In the literature, first attempts and a practical implementation to reduce this effect by signal diversity can be found. In this article, a practical implementation with four measuring heads of a Multipoint Vibrometer (MPV) and an evaluation and optimization of an algorithm from the literature is presented. The limitations of the algorithm, which combines velocity signals, are shown by evaluating our measurements. We present a modified algorithm, which generates a combined detector signal from the raw signals of the individual channels, reducing the mean noise level in our measurement by more than 10 dB. By comparing the results of our new algorithm with the algorithms of the state-of-the-art, we can show an improvement of the noise reduction with our approach.

## 1. Introduction

With the advancement of laser-Doppler vibrometers (LDVs), various additional applications are continuously made available in which contactless vibration measurement is possible [1,2,3]. These applications are difficult to implement with conventional methods of vibration measurement, impossible in the case of rotating or hot parts [4], or unwanted in medical applications [5]. 

Throughout this article, we use laser-Doppler vibrometer (LDV) to refer to heterodyne LDVs, such as described in [6,7]. One limitation in LDV measurements is the impact of the laser speckle effect [6,7,8,9], which leads to so-called signal dropouts [7] on rough or fast-moving surfaces [10].

In this case, no light reaches the photodetector at a certain point in time and consequently no information about the vibration can be obtained. The impact of this effect is the subject of several publications and has already been widely researched [9,10,11,12,13]. Even though the effect can be utilized for some specific measurement methods [14,15], in most LDV applications, it reduces the signal quality and limits the minimally detectable amplitude of a vibration [7]. Therefore, a reduction of the impact of this effect is desirable. 

Various approaches using adaptive optics have been used to accomplish this [16,17,18]. For this purpose, however, commercial vibrometers must be modified to a large extent. Another method to achieve this is through signal diversity, which is widely used in radio communications [19,20]. With signal diversity, the signal is detected from multiple channels. The fundamental idea of improving the signal quality through signal diversity is the assumption of stochastically independent signal dropouts of the individual channels, caused by the laser speckle effect. Therefore, the probability of a signal dropout occurring on all channels at a given time is exponentially lower with an increasing number of channels [7,21].

First results to improve the signal quality through signal diversity have been published by Dräbenstedt [18,21]. In his publications, an algorithm for calculating a combined signal from two or more demodulated velocity signals was developed and subsequently tested in a practical experiment.

This article first aims to confirm Dräbenstedt’s results by an experimental verification using four channels of a Multipoint Vibrometer (MPV). For our purposes, the individual measuring heads (channels) of the MPV can be seen as independent conventional LDVs and therefore all results can be achieved with several conventional LDVs in the same way. 

The limitations of the algorithm are shown by evaluating our measurements. Subsequently, we derive a modified algorithm, which generates a combined signal from the raw signals of the individual channels and compare the results of the algorithms. By this comparison we can show the improvement of the resulting combined signal with our modified algorithm. 

We would like to mention at this point that we do not require real-time performance if the result is obtained within a reasonable amount of time. We are focusing on investigating whether an improvement of the signal quality is possible, so a slightly increased processing time is not relevant at this time.

## 2. Materials and Methods

To verify the results obtained by Dräbenstedt [18,21], an experiment using four channels (measuring heads) of a MPV is conducted. The measuring heads are aimed at a shaker with a nearly identical angle of incidence. To verify the alignment of the measuring heads, the raw signals of the four channels are acquired and subsequently demodulated by an ATAN demodulation [22]. A correct alignment can be verified by a match of the four demodulated velocity signals. We aimed for a difference of less than 5% between the amplitudes of the four velocity signals. 

According to Dräbenstedt’s publication, the signal reliability can be improved by combining multiple demodulated velocity signals. Using Equation (1), the combined signal S is obtained using any number n of individual velocity signals $X_{j}$
(1)$$S=\overset{n}{\underset{j=1}{\sum }}w_{j}X_{j}$$
with the weighting factors $w_{j}$ calculated in Equation (2) [21] by the CNR (Carrier to Noise Ratio)
(2)$$w_{j}=\frac{1}{\sum_{i=1}^{n}\frac{CNR_{i}}{CNR_{j}}}.$$

The CNR results from Equation (3) with the carrier power $P_{c}$ and the noise power $P_{n}$.
(3)$$\mathrm{CNR}=10{\;\log}_{10}\frac{P_{c}}{P_{n}}$$

The carrier and the noise power can be calculated with an estimation of the spectral power density with the MATLAB™ function *periodogram*. For calculating the carrier power $P_{c}$, for short signal lengths of 1000 samples, we assume a bandwidth of 600 kHz around the carrier frequency of 2.5 MHz. The noise power $P_{n}$ is calculated from the power of the remaining frequency band (bandwidth depending on the sampling rate Fs). 

The following experiment is intended to generate suitable velocity signals, for testing the algorithm based on Equations (1) and (2) [21]. Our goal is to generate an artificial and easy to replicate signal dropout by disrupting the beam paths to a shaker, which is used as a source for a known vibration. We achieve this with a rotating disc with holes for letting the beams pass through. Due to the rough surface of the disc and its placement out of focus of the laser beams, very little light is reflected, and a signal dropout is forced. This experimental setup, where the rotating disc is moved by a stepper motor, is shown in Figure 1. 

The four beams are aligned to ensure that at least one of the laser beams of the channels CH1–CH4 always passes through the holes and is positioned on the shaker. The results for this experiment are discussed in Section 3. 

Following the initial experiment with an artificial signal dropout, a more realistic experiment is performed. For this purpose, the four channels of the MPV are focused to the same spot on a test object. Unlike the previous experiment, only one of the measuring heads actively generates a laser beam (active channel), whereas the other three measuring heads only receive scattered light (passive channels). The experimental setup is shown in Figure 2. 

For this experiment, the measuring heads are aligned on a vibrating speaker with a known frequency. Due to the angle of the laser beams and the vibration of the speaker, in-plane movement also occurs, resulting in noise caused by the laser speckle effect. With both experimental setups, several measurements are acquired and analyzed. The following section describes the results of our evaluation.

## 3. Results

### 3.1. Results from the First Experiment Using Dräbensted’s Algorithm for a Combined Signal

The four signals of a measurement obtained from the first experimental setup, as shown in Figure 1, are demodulated and the resulting velocity signals are displayed in Figure 3. In addition, the combined signal derived from the velocity signals of CH1–CH4 with Equations (1) and (2), is also pictured. 

In Section 3.2, a detailed explanation of the implemented algorithm for raw signals can be found, which is applicable to velocity signals as well. 

For the channels CH1–CH4, strong peaks (due to signal dropouts) are visible in the velocity signals. In our application with forced signal dropouts, we know that at any time a channel exists, where a signal without any disturbances can be detected. This fact can generally be assumed for any signal, as the signal dropouts caused by the laser speckle effect are stochastically independent [21].

We can confirm this by examining the combined signal (VeloComb from [21] in Figure 3). In contrast to the individual velocity signals (CH1–CH4), the vibration of the shaker at 100 Hz is clearly visible in the combined signal. 

The functionality of the algorithm can also be shown in the spectral results of this measurement shown in Figure 4. It should be mentioned that the detected frequency at 100 Hz has approx. The same amplitude for all signals; however, the noise level of the combined signal is significantly lower.

#### 3.1.1. Limitations of the Algorithm

Upon closer inspection of the velocity signal of the combined signal (VeloComb), some smaller noise peaks can still be seen. For a better illustration, a magnified section of the velocity signals from Figure 3 are shown in Figure 5. 

The cause of these peaks can be explained by examining the weighting factors $w_{j}$ from Equation (2), required for the determination of the combined signal. We calculate these with a section of the signal with a length of 1000 samples. In the section shown in Figure 5, the weighting factors are $w_{1}=0.8,\;w_{2}=0.11,\;w_{3}=0.05$ and $w_{4}=0.04$. Thus, CH1 has the greatest influence on the combined signal at 80%. Considering the corresponding section of the raw signal, shown in Figure 6, this estimation is realistic.

Considering Figure 5 and Figure 6, the cause of the peaks of the combined signal can be attributed to the voltage drop of CH2, which is illustrated with the upper signal envelope of the raw signals, shown in Figure 6b. Despite a small weighting factor for CH2, this causes a large peak in the resulting combined velocity signal. 

An additional source of error are the transition points of the sections where discontinuities can occur. This will be discussed in more detail further in the paper.

To account for the error of the voltage drop from CH2 (possibly caused by a signal dropout), $w_{2}$ needs to be close to zero at the displayed section. For this purpose, the time interval used for deriving the weightings factors must be shorter than approximately 5 s. Depending on the sampling rate (we recorded 10 MSamples and sampled with either 25 MHz or 10 MHz, which is both sufficient for the carrier frequency of 2.5 MHz), this corresponds to 200 or 80 samples for the determination of the CNR to calculate the weighting factors $w_{j}$. 

In addition to the significantly increased computing cost, the susceptibility to errors of the calculated CNR is significantly higher with fewer samples. This in turn can lead to further errors that cannot easily be compensated. Therefore, the signal quality is only slightly better even with a short sample length for determining the weighting factors. 

A possible solution to this problem is to use a considerably higher sample rate, which leads to an exponentially higher computing cost and, therefore, is not possible. 

An easier to implement method to prevent these errors is to introduce an exponent in the calculation of the weighting factors. This still only works in certain cases and is implemented in Equation (6) in the following section, which describes a modified algorithm developed by us. 

Another approach for signal optimization is the combination of the raw signals of the individual channels instead of the velocity signals. However, there is a non-constant phase difference of the individual channels, which prevents a simple addition of the raw signals [21]. Therefore, a simple addition of the signals can lead to an elimination of the combined signal. This is illustrated in Figure 7, showing the combined signal calculated from the raw signals from the first experimental setup, shown in Figure 1, based on the weighting factors from Equation (2).

At the magnified section (top right in Figure 7), the combined signal is obtained from equal parts of CH2 and CH4 by the weighting factors. As these channels have a similar amplitude and are shifted by approx. 180° in their phase, the resulting signal is close to zero. 

Overall, the algorithm from Dräbenstedt [21] for calculating the combined signal does still yield very good results, as shown in Figure 3 and Figure 4. In the following sections we attempt to further improve the results of the combined signal, to achieve an even better signal reliability. 

### 3.2. Modified Algorithm to Obtain the Combined Signal from Raw Signals

In order to solve the discontinuity problems at the transition points of the sections, an examination of these transition points is necessary. In this section we examine, if the problem of the non-constant phase difference of the raw signals of the individual channels with respect to each other can be solved simultaneously. 

For this purpose, the individual channels are digitized with a sampling rate Fs (either 10 or 25 MHz). Each digital signal consists of N samples (here 10 MSamples) and is subsequently split into blocks with a length of k (here 1000 samples). As mentioned above, a much shorter block length requires a lot of computing power and causes problems in the reliable determination of the CNR for the weighting factors. This process is illustrated for a channel in Figure 8. 

The weighting factors are then calculated for the resulting blocks of the four channels depending on the signal strength. These weighting factors can be calculated either in the time or frequency domain (for calculating the CNR). For the calculation in the time domain, an auxiliary factor is calculated from the median of the absolute value of the block of a signal according to Equation (4).
(4)$$A_{i}=\mathrm{median}\left(\left|{\mathrm{CH}}_{i}\left(1:k\right)\right|\right)$$

Alternatively, this factor can be determined in the frequency domain according to Equation (5) with the CNR, which is calculated
(5)$$A_{i}=\mathrm{CNR}\left({\mathrm{CH}}_{i}\left(1:k\right)\right)$$

The factors A which are proportional to the signal strength, are then normalized so that their sum is one. The resulting weighting factors $F_{\mathrm{CH},i\;}$, for one block of the length k, are thus calculated by Equation (6).
(6)$$F_{\mathrm{CH},i\;}=\frac{A_{i}^{\alpha }}{\sum_{i=1}^{4}A_{i}^{\alpha }}$$

The exponent α allows a stronger weighting to be implemented. For large α the weighting of the channels with greater signal strength in the combined signal is exponentially increased (for $\alpha \to \infty ,\;\max\left(F_{\mathrm{CH},i}\right)=1$). For the calculation via the CNR and for α = 1, these factors are equivalent to the factors calculated by Equation (2). For α > 2 the shown problem in Figure 5 can already be decreased significantly. The combined signal can then be calculated blockwise from the resulting weighting factors $F_{\mathrm{CH},i}$.

Altogether, the combined raw signal ${\mathrm{CH}}_{\mathrm{combined}}$ is given by Equation (7), with N total samples split into m=N/k blocks, with a length of k samples each (in this case m=10,000).
(7)$${\mathrm{CH}}_{\mathrm{combined}}\left[k\cdot \left(j-1\right)+1:k\cdot j\right]=\overset{4}{\underset{i=1}{\sum }}{\mathrm{CH}}_{i}\left[k\cdot \left(j-1\right)+1:k\cdot j\right]\cdot F_{\mathrm{CH},i\;}\left(j\right),\;\;\;\;j=1\ldots m$$

The result of the calculation using Equation (4) or Equation (5) differs only slightly, as both methods have a similar proportionality to the signal strength. For future work we will consider their computing times. 

As mentioned above, one problem with this calculation are discontinuities at the combined signal blocks, which contribute to a distortion of the signal and to a higher noise level. Furthermore, there is a time-invariant phase offset between the individual raw signals, which, as shown in Figure 5, can lead to an elimination as well as other incorrectly detected frequencies. To solve this, we implemented an algorithm that shifts the blocks of the individual channels in phase by means of peak detection, to match their phase. The algorithm first finds the peaks ${\vec{P}}_{i,j}$ of the sinusoidal raw signals by Equation (8). We implemented this with the MATLAB^TM^ function *peaks*, but a customized implementation via a local maxima detection is also possible.
(8)$${\vec{P}}_{i,j}=\mathrm{peaks}({\mathrm{CH}}_{i}\left[k\cdot \left(j-1\right)+1:k\cdot j\right],\;\;\;\;\;\;i=1\ldots 4,\;j=1\ldots m$$

Then the sample length of the phase offset $a_{i,j}$ is calculated in Equation (9) by using the first detected peak $p_{i,j}={\vec{P}}_{i,j}\left(1\right).$
(9)$$a_{i,j}=p_{i,j}-\min\left(p_{1,j},p_{2,j},p_{3,j},p_{4,j}\right),\;\;\;\;\;\;i=1\ldots 4,\;j=1\ldots m$$
with the calculated sample length of the phase offset $a_{i,j}$ the individual raw signals are phase shifted according to Equation (10).
(10)$${\mathrm{CH}}_{i,\mathrm{PS}}\left[k\cdot \left(j-1\right)+1:k\cdot j\right]={\mathrm{CH}}_{i}\left[k\cdot \left(j-1\right)+1+a_{i,j}:k\cdot j+a_{i,j}\right],\;\;\;i=1\ldots 4,\;j=1\ldots m$$

An example of a small section of one block of the raw signals before and after correction, is shown in Figure 9. 

In the selected section of the raw signal, the first detected peak amplitude belongs to CH1, consequently all other channels are aligned accordingly. In this case CH2 is shifted by three samples, CH3 by five samples and CH4 by eight samples.

Afterwards, the samples before and after the transition points are interpolated to correct missing samples due to the phase shift and discontinuities due to the blockwise combination. 

In Figure 10, a small section of the combined demodulated velocity signal around a transition point of two blocks (between j = 1 and j = 2 with k = 1000) is shown before and after interpolation. 

Due to this preceding method, we can minimize the impact of both the discontinuities and the phase offset without distorting the original signal, as can be seen by the magnified section in Figure 10. 

This is possible, because these errors always occur in the same locations around the transition points. Errors similar to the one shown in Figure 5 are random and therefore much harder to compensate.

### 3.3. Comparison of the Algorithms

With our modified algorithm a combined velocity signal is calculated (with α = 5) from the same measurement and from the first experimental setup as before. The combined signal from the raw channels, the demodulated velocity signals of the individual channels and the combined signal from the velocity signals is shown in Figure 11.

The resulting velocity signal from our algorithm is significantly less noisy. The peaks of the combined signal, in comparison to the combined signal derived from the demodulated velocity signals, are mostly gone. Figure 12 additionally shows the displacement signals derived from the velocity signals. The offset resulting from the difference between the actual carrier frequency and the carrier frequency assumed for demodulation was compensated for each signal.

The functionality of the algorithms is evident in both combined signals—because most of the time just one channel is affected by signal dropouts, this time segment can be replaced in the combined signal by the other channels.

The shaker’s vibration frequency of 100 Hz is visible in all signals. For the combined signal from the old algorithm, a slight offset can still be seen compared to the combined signal from the algorithm developed by us. The cause of this offset can be explained by the disturbances visible in the velocity signal in Figure 11. 

The difference between the algorithms can also be shown by the significantly lower noise in the frequency spectrum, as shown in Figure 13.

The relatively large amplitude of the shaker at 100 Hz is detected with a similar amplitude in all four channels as well as in the combined signals. Because of the signal dropouts, forced by our setup, in the individual channels, the noise level is significantly higher. 

For CH1 and CH2, the amplitude at 100 Hz is just slightly above the noise level, making the detection of an unknown frequency unrealistic. Generally, only higher amplitudes can reliably be detected in the individual channels, that are affected by signal dropouts.

In order to estimate the noise reduction more accurately, Figure 13b shows the smoothed frequency spectra of the combined as well as one individual velocity signal. For this measurement, the algorithm, that calculates the combined signal from the velocity signals (VeloComb), reduces the mean noise level in the frequency range up to 5000 Hz by 17 dB. Our algorithm that calculates the combined signal from the raw signals (RawComb) decreases the mean noise level by an additional 15 dB. Therefore, only the algorithm using the raw signals will be described in the following parts of this article, as it consistently achieves better results.

All previously shown results used measurements from the first experimental setup, which had the main objective of generating a reliable, reproducible signal as a basis for developing and testing the presented algorithm. To test our algorithm further, the results from the second experiment with only one active channel, which is much closer to real world applications, are presented in the following section. 

#### 3.3.1. Further Examination of the Developed Algorithm through the Second Experiment

The first experiment results in predictable signals that are helpful for the development of the algorithm. The signals are thus applicable to real world applications in a limited extent only. For a more representative comparison, the second experiment is more suitable. Using the experimental setup for the second experiment shown in Section 2, measurements are recorded and demodulated (Fs = 10 MHz, RBW = 1 Hz). In Figure 14, the demodulated velocity signals of the four channels as well as a combined signal derived from the four raw signals of one measurement are shown. By aiming the measuring beam at a poorly reflecting part of the speaker, the signal level is relatively low and signal dropouts can be seen in the demodulated signal.

Compared to the measurement from the first experimental setup on the shaker and four active channels, the measurement on the speaker and only one active channel results in a considerably higher noise level in the velocity signals.

For the active channel CH4 and the passive channels CH1–CH3, numerous signal dropouts can be seen. Since the signal dropouts are uncorrelated, it is likely that one channel detects the vibration of the speaker at any given time. The fundamental vibration of the speaker at 205 Hz is recognizable in the displacement signals derived from the velocity signals, shown in Figure 15.

The disturbances of the individual channels are also visible in the displacement signals, resulting in a higher noise level of the frequency spectra, shown in Figure 16.

Due to the lower noise level of combined signal, the functionality of the algorithm can be shown. The relatively large amplitude of the speaker’s vibration frequency is detected almost identically by all signals. With these results, we can demonstrate that the algorithm yields good results for close to real-world applications. For this measurement, the mean noise level in the frequency range up to 5000 Hz of the combined signal (RawComb) was reduced by 10 dB compared to the mean noise level of the signals of the individual channels.

#### 3.3.2. Conditions for Successful Diversity Measurements

For a successful measurement, a correct alignment of the passive measuring heads is essential, otherwise insufficient amounts of light reach the detectors resulting in a correspondingly poor signal quality. Figure 17 shows an example of such a case. 

For this purpose, the measurement object is arbitrary since the aim of this section is only to demonstrate a measurement with incorrect alignment. For this measurement, the speaker was removed, and the measurement was conducted on the laboratory table underneath (no active vibration). The resulting poor alignment of the measuring heads results in a poor signal quality for all passive channels, as very little light reaches the sensors of the measuring heads. This causes numerous peaks in the velocity signals of CH1 and CH2. In such a case, the combined signal obtained by our algorithm is largely equivalent to the signal with the highest signal strength (in this case CH4). This is visible in the matching velocity signal in Figure 17 and in the almost identical frequency spectrum, shown in Figure 18.

## 4. Discussion

With the first experiment, we verified the functionality of the algorithm developed by Dräbenstedt [21] for deriving a combined signal from four velocity signals. By examining the resulting combined signal, we identified sources of potential errors, which limit the signal quality of the combined signal which has already been significantly improved. 

We derived an algorithm that calculated a combined signal based on four raw LDV signals. Subsequently, both algorithms were applied to the same measurement and revealed that the newly developed algorithm allows an improvement of the combined signal. 

For measurements with the first experimental setup, we were able to show that the mean noise level of the combined signal is reduced by an additional 15 dB (RBW = 1 Hz) in the frequency range up to 5000 Hz, when calculated with our algorithm (RawComb), compared to the algorithm from the literature (VeloComb), as shown in Figure 13b. For the second experimental setup the additional reduction was approx. 6 dB (RBW = 2.5 Hz). 

Based on the evaluations of the first and second experiments, we demonstrated the basic functionality of the developed algorithm for determining a combined signal from the raw signals of multiple channels. 

Specifically, it was shown that the combined demodulated signal is at least as good as the signal of the best individual channel. Depending on the disturbances and signal dropouts that occur, the signal quality, and accordingly the noise level, of the combined signal can be significantly better than that of a signal from a single channel. In the results of our measurements of the second experimental setup, as shown in Figure 16, a mean noise reduction in the frequency range up to 5000 Hz of more than 10 dB was achieved (RBW = 2.5 Hz). 

The results seem plausible and support the findings of [18,21], as both algorithms for determining a combined signal significantly reduced the noise level caused by signal dropouts due to the laser speckle effect. The increased reduction of the noise level (up to 6 dB in the second experiment) by our algorithm (RawComb) compared to the algorithm from the literature (VeloComb) is reasonable, since our algorithm does not have the limitations shown in Section 3.1.1.

The implementation of our findings could improve the signal quality in various applications involving measurements on rough or moving surfaces, where the signal quality is reduced by laser speckle effects and the resulting signal dropouts [7,10]. 

A possible use case is medical applications, where LDVs are used for contactless measurements, such as monitoring cardiovascular activity [5,23]. In such cases, it is not always possible to guarantee sufficient reflectivity of the skin and that the patient does not move, or the appropriate procedures involved in ensuring this require greater effort [23]. For this application, signal diversity could decrease noise contribution of laser speckle effects, or even eliminate the need for time-consuming preparation of the skin.

For future research we are investigating additional real-world applications as well as the real-time capabilities of our algorithm and the impact of implementing peak-filtering algorithms in the individual signals before combining them. Furthermore, the question of to what extent the measurements can be compensated with respect to the angle of incidence needs to be addressed in order to obtain reliable and accurate measurements in real-world applications.

## Figures and Tables

**Figure 1 sensors-21-00998-f001:**
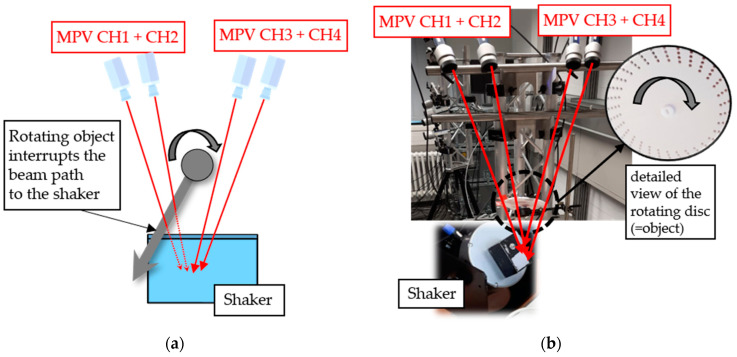
(**a**) Concept of the experiment to obtain velocity signals with signal dropouts; (**b**) Image of the actual test setup; laser beams go to the shaker in a straight line through the holes in the rotating disc.

**Figure 2 sensors-21-00998-f002:**
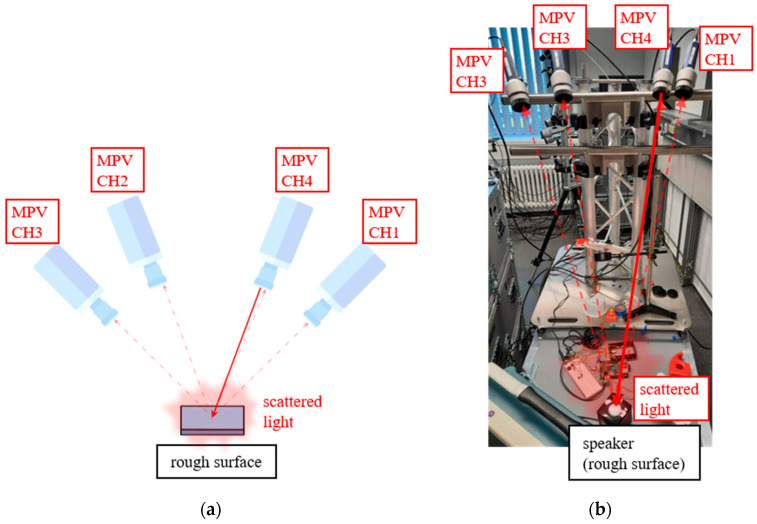
(**a**) Concept of the second experimental setup with one active measuring head (CH4) and three passive measuring heads (CH1–CH3); (**b**) Image of the actual test setup measuring on a speaker.

**Figure 3 sensors-21-00998-f003:**
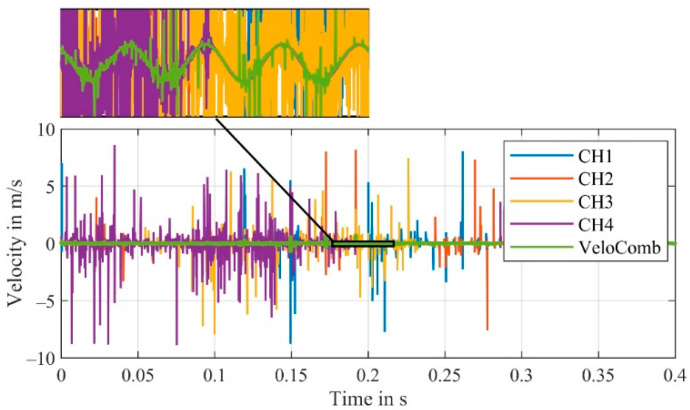
Velocity signals of the individual channels (CH1–CH4) and the combined velocity signal (VeloComb.); (RBW = 2.5 Hz); additional detailed view of a small section to better show the combined signal.

**Figure 4 sensors-21-00998-f004:**
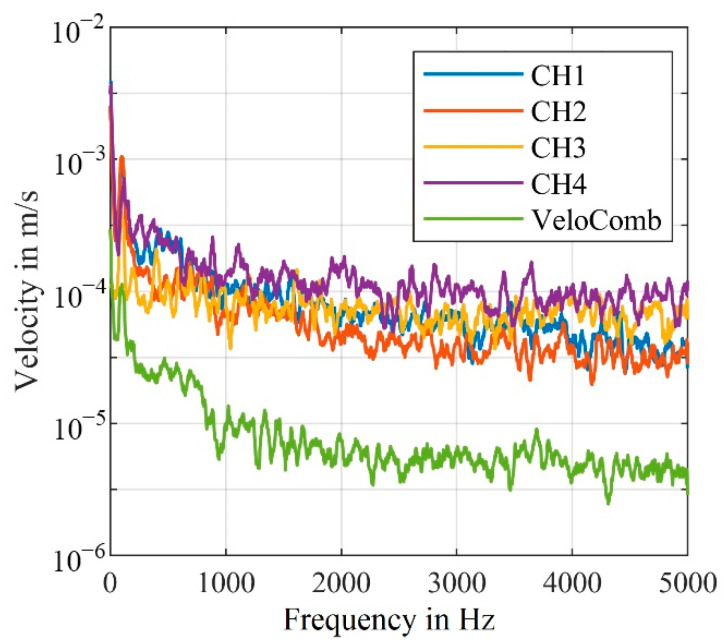
Frequency spectra of the velocity signals shown in Figure 3 (smoothed for better visualization); (RBW = 2.5 Hz).

**Figure 5 sensors-21-00998-f005:**
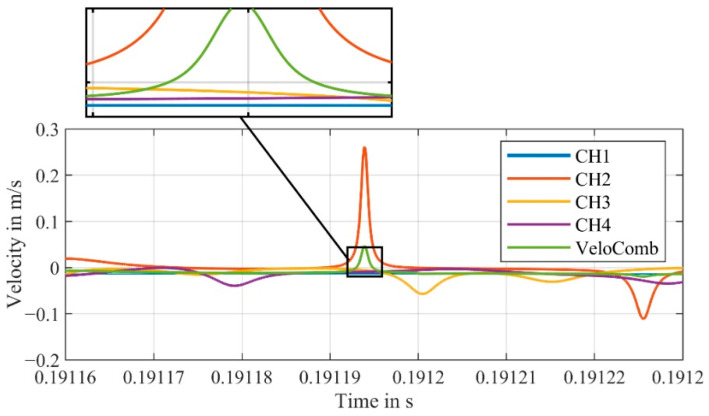
Detailed view of the velocity signals of the individual channels (CH1–CH4) as well as the combined signal (VeloComb); (RBW = 2.5 Hz).

**Figure 6 sensors-21-00998-f006:**
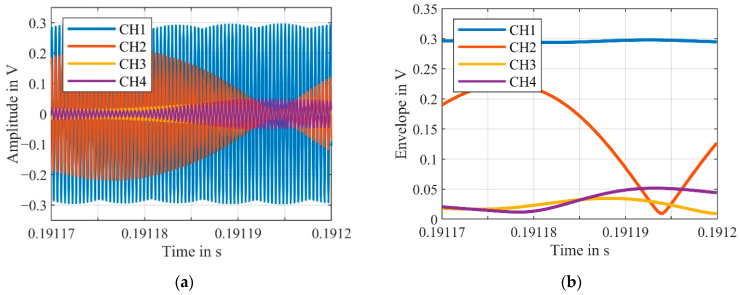
(**a**) Raw signals of the individual channels (CH1–CH4) at the same time segment as the velocity signals shown in Figure 5; (**b**) Upper signal envelope of the raw signals.

**Figure 7 sensors-21-00998-f007:**
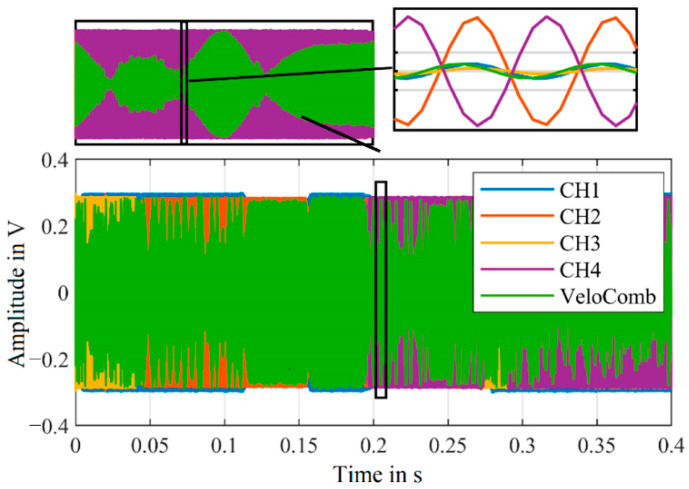
Raw signals of the individual channels and the combined signal (VeloComb) derived from the individual channels and weighting factors from Equation (2).

**Figure 8 sensors-21-00998-f008:**
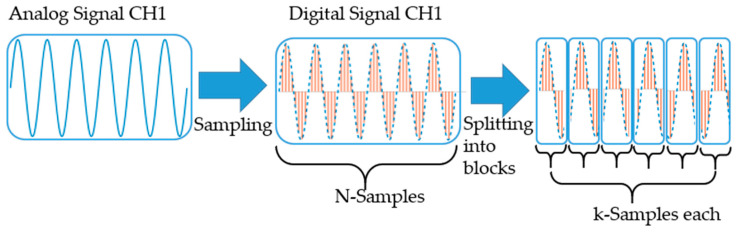
Visualization of the process of digitizing and splitting the individual channels into blocks (the shown signal is chosen only for visualization with a low frequency; in the real signal there are several hundred periods in each block).

**Figure 9 sensors-21-00998-f009:**
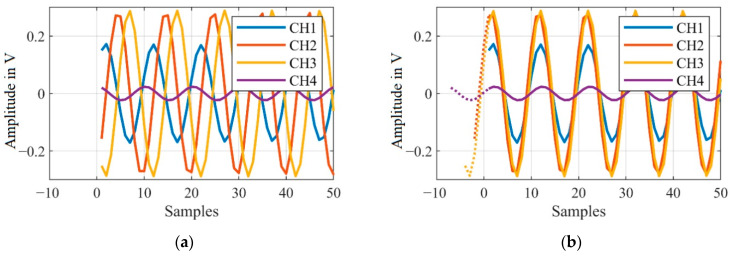
(**a**) Raw signals of the individual channels (CH1–CH4) without phase correction; (**b**) Raw signals with phase correction (samples with dotted lines are only shown for visualization of the shift).

**Figure 10 sensors-21-00998-f010:**
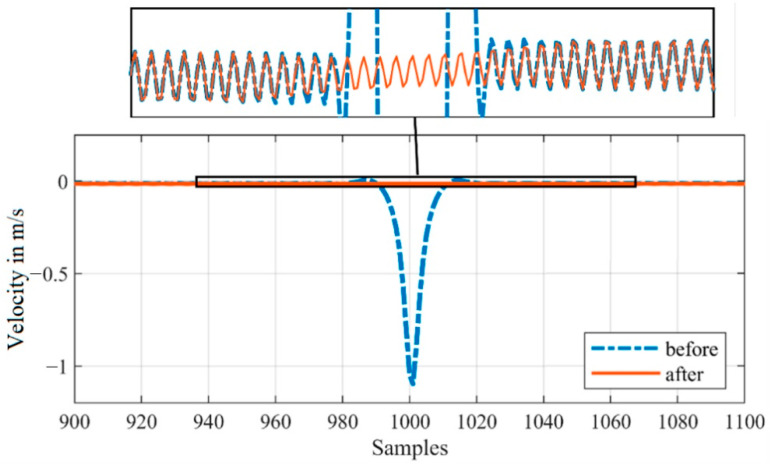
Section of the combined, demodulated velocity signal before and after interpolation to correct discontinuities and error due to phase shifting.

**Figure 11 sensors-21-00998-f011:**
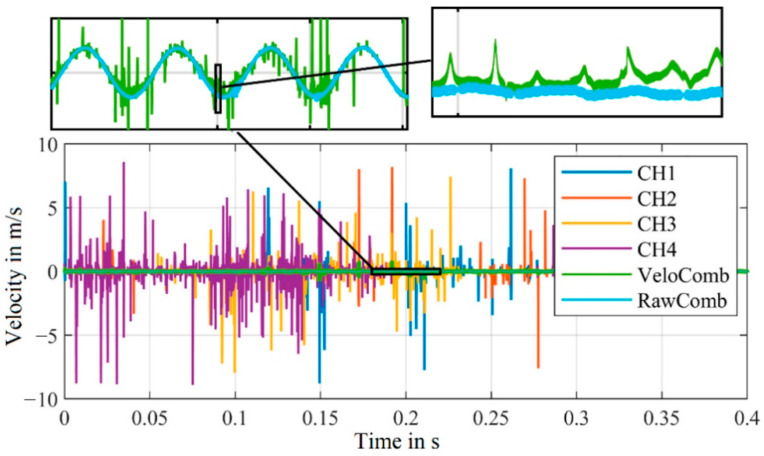
Comparison of the demodulated velocity signals of the channels CH1–CH4 and the combined signals (from [21] “VeloComb” and from our algorithm “RawComb”); zoomed sections only for combined signals to illustrate less noise; (RBW = 2.5 Hz).

**Figure 12 sensors-21-00998-f012:**
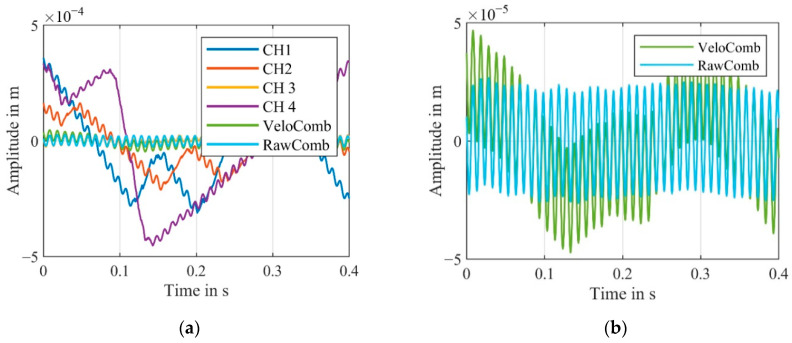
(**a**) Comparison of the demodulated displacement signals of the four channels CH1–CH4 as well as the combined signals derived from the old and new algorithms; (RBW = 2.5 Hz). (**b**) Magnified section of the combined signals from (**a**) to illustrate the difference.

**Figure 13 sensors-21-00998-f013:**
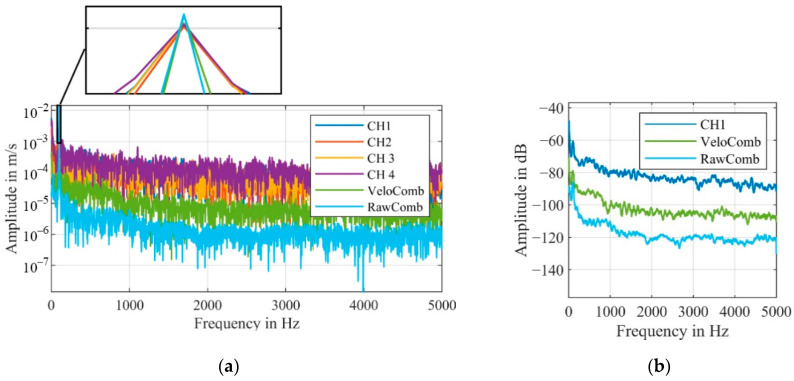
(**a**) Frequency spectra of the velocity signals with a detailed view of the vibration frequency at 100 Hz; (RBW = 2.5 Hz). (**b**) Smoothed frequency spectra for the combined and one individual velocity signal with the amplitude in dB (0 dB = 1 m/s); same scale as in (**a**).

**Figure 14 sensors-21-00998-f014:**
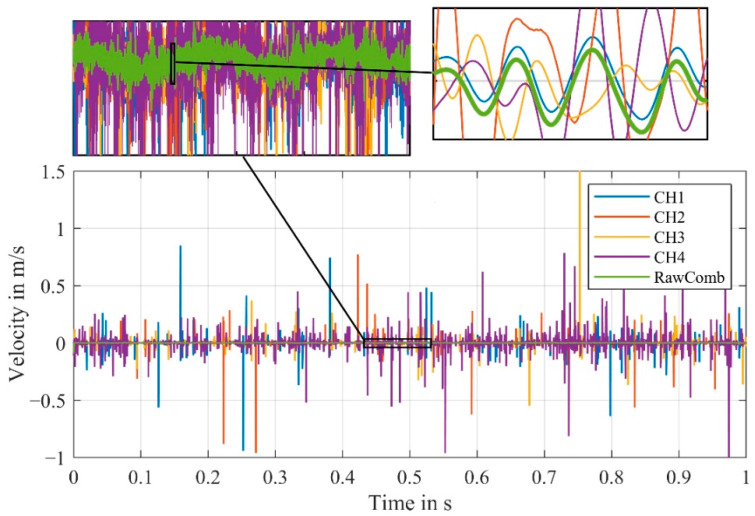
Comparison of the demodulated velocity signals of the four channels CH1–CH4 and the combined signal from the raw signals (VeloComb); (RBW = 1 Hz).

**Figure 15 sensors-21-00998-f015:**
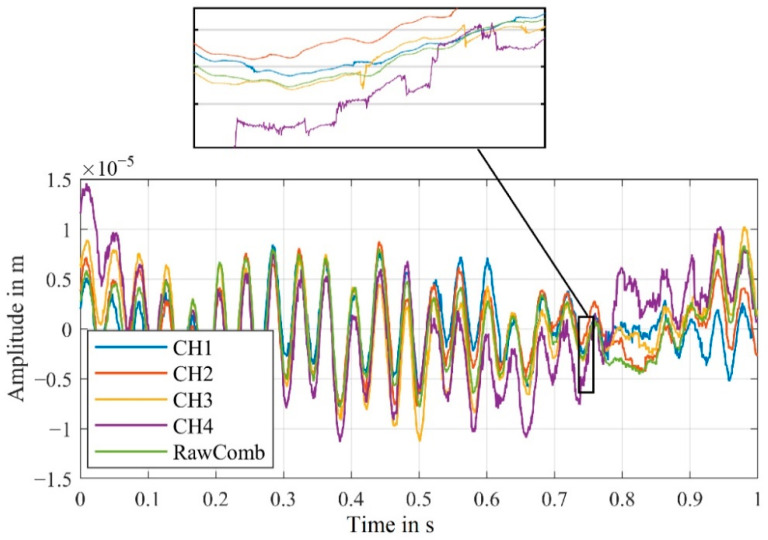
Comparison of the demodulated displacement signals of the four channels CH1–CH4 and the combined signal; (RBW = 1 Hz).

**Figure 16 sensors-21-00998-f016:**
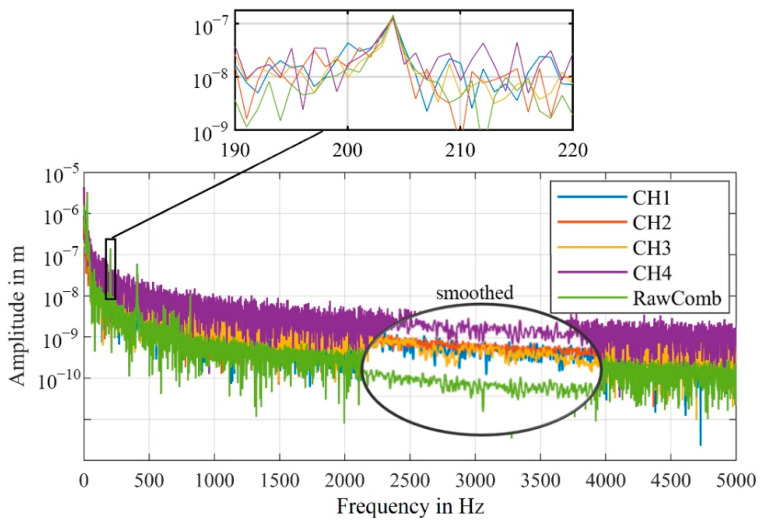
Comparison of the frequency spectra of the four channels CH1–CH4 and combined signal, one section of the spectra is shown smoothed for better visualization; (RBW = 1 Hz).

**Figure 17 sensors-21-00998-f017:**
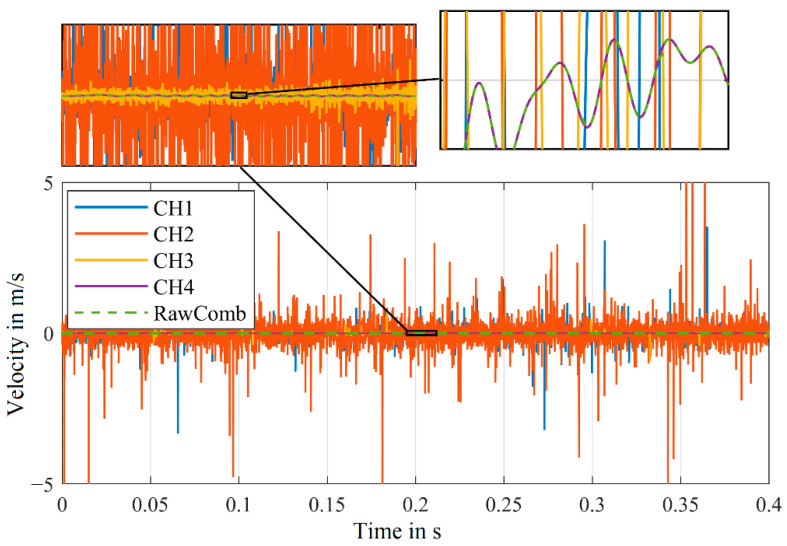
Velocity signal of the four channels CH1–CH4 and the combined signal with poor alignment of the measurement heads; (RBW = 2.5 Hz).

**Figure 18 sensors-21-00998-f018:**
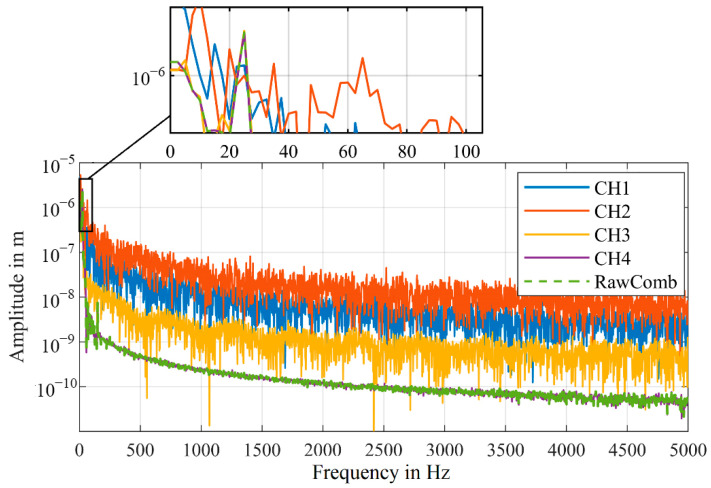
Frequency spectra of the four channels CH1–CH4 and the combined signal; poor alignment of the measurement heads; (RBW = 2.5 Hz).

## Data Availability

The raw data analyzed in this study is available online at: [www.iei.tu-clausthal.de/ueber-uns/veroeffentlichungen/daten].

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
