# Peer review of "Signal Diversity for Laser-Doppler Vibrometers with Raw-Signal Combination"

_sensors, 2021, doi:10.3390/s21030998_

Round 1

Reviewer 1 Report

This is an interesting and well written paper on the important topic of how to overcome signal dropout in laser vibrometry measurements. I have a couple of broader comments for the authors and some minor comments that I think they should deal with to improve the paper.

Broader comments:

In practice, for a target with arbitrary motion, it would be impossible or at least less easy to compensate the measurements correctly for angle of incidence. Would this limit the applicability of this method for real applications or is there a way around the problem?

For further work, I would encourage the authors to use a target with a broadband vibration and make a quantitative analysis of errors in amplitude and phase between the RawComb measurement and an independent true measurement.

Detailed comments:

I encourage the authors to check mathematical notation in equations. For example, the use of X in equation 1 differs from its use in equations 4 and 5. Also, the use of counters i and j may not always be consistent (see also line 224) and I wasn't sure that all the variables had been declared in equation 7.

is equation 2 obvious or does it need a reference?

I didn't find figure one particularly clear.

Most of the analysis in the paper is based around the experiment in which there are four active channels with a rotating disc used to chop the light at different times for each channel. The second experiment in which there is only one active channel receives much less attention in the paper, just a short description in Section 3.3.1. I wasn't clear what additional point was being made in Section 3.3.1 so the authors may want to emphasise the point if I've missed it or may reconsider the inclusion of this second experiment. I was particularly unclear about the point being made in the very last paragraph of Section 3.3.1 where the measurement is made on the stationary laboratory table.

Just before figure 3, the authors say that “a detailed explanation of the implemented algorithm can be found in section 3.2” but section 3.2 is about the addition of raw signals rather than the addition of velocity signals.

At the top of page 6, the authors seem to rule out an approach to address the deficiencies of the original approach because it requires a higher sample rate but then adopt an alternative approach that also requires a similarly high sample rate. I may have misunderstood this.

On line 167, should it say 180 degrees?

In Figure 8, the sampling looks to be irregular and I didn't understand why.

Should equation 5 make clear that the CNR operation is performed on a Fourier transform of the raw signal rather than on the raw signal itself?

In equation 6, it seemed to me that the ratio would always be less than one so I did not understand why the maximum value of F would be 1 if alpha was infinite.

In Figure 9, should the vertical axes be labelled as amplitude in V rather than m/s?

in Figure 10, should the vertical axis be labelled as velocity rather than amplitude to be consistent with, for example, figure 5?

Line 274 makes reference to a chapter . Should this be a section?

Author Response

Thank you very much for your detailed review! Your comments were very helpful. Attached you will find our responses to the comments. The implemented changes can be found in the new manuscript.

Reviewer 2 Report

The manuscript titled “Signal Diversity for Laser-Doppler Vibrometers with Raw-Signals Combination” reports an interest measurement topic of Laser Doppler Vibrometry. The research work is described in enough detail and the work seems quite solid. Therefore, I feel that the paper should eventually be published. However, before the paper is ready for publication, there are a couple of issues that should be addressed:

  • The main focus of the paper is related to the Laser Doppler Vibrometry technique but very little is said about the principles of the technique. The authors of the paper can add an additional section in the introduction of the manuscript explain the LDV principles and also the LDV applications.
  • Regarding the LDV applications, the work presented by the authors of the paper is an important topic that can be better address in the discussion section. The authors can explain how this new algorithm can be used in real applications to reduce the noise during measurement on rough or fast-moving surfaces. The authors can expand the discussion section also explaining the possible field of application of this new algorithms, as for example, in the medical scenario where a user is moving or the skin is not well prepared. For example, in medical scenario, not always is possible to clean the skin, use a retroreflective treatment or be sure that the user is not moving, in particular during tests like this:

Casaccia, S., Sirevaag, E. J., Richter, E. J., O’Sullivan, J. A., Scalise, L., & Rohrbaugh, J. W. (2016). Features of the non-contact carotid pressure waveform: Cardiac and vascular dynamics during rebreathing. Review of Scientific Instruments, 87(10), 102501.

  • Numerical data of the reduction of the noise using the algorithm from the literature and the new one proposed by the authors is missing. It is difficult to see that difference from the graphs reported in the manuscript (fig 11, 12 and 13). Also in the discussion section, the authors can report the improvement reached through the new algorithm in confront of the algorithm developed by Dräbenstedt.

Author Response

We appreciate your review! Attached you can find our responses to the comments. The implemented changes can be found in the new manuscript.

Reviewer 3 Report

To the authors:

Hereby, my comments to the paper: “Signal Diversity for Laser-Doppler Vibrometers (LDV) with Raw-Signal Combination”.  The main idea of this paper, is to overcome drawbacks of the LDV system, when the measurement is performed in a rough surface. The photodetector could have difficulties to detect the light, when is scattered and cannot reach fully. The low intensity of the reflected laser beam leads to a signal dropout, which establishes as noise peaks in the demodulated velocity signal. In such circumstances, no light reaches the detector at a specific time and, it cannot be detected. To solve this, it was implemented four measurement sensors in order to overcome the above difficulties. It was also developed a new algorithm.

.

My comments in details:

·         The obtained results have to be quantify in the abstract and conclusions part.

·         The authors need to perform and extensive validation process, to their findings.

·         How was the correct alignment of the measurement heads performing, and how the precision values were?

·         It was not clear how the reduction of signals discontinuities were done

·         The authors stated that the finding is especially relevant for longer measurements and measurements on objects with a rough surface. As I can see it was not properly verified this situation in the paper.

Author Response

Thank you for your useful comments! Attached you can find our responses to the comments. The implemented changes can be found in the new manuscript.

Round 2

Reviewer 2 Report

The authors answered to all the reviewer's comments.

Reviewer 3 Report

Dear Authors:

I got through the new version of the paper . You addressed my comments and it can be accepted for publication